# Learning multiple visual domains with residual adapters

**Sylvestre-Alvise Rebuffi**[1]    **Hakan Bilen**[1,2]    **Andrea Vedaldi**[1]

[1] Visual Geometry Group
University of Oxford
{srebuffi,hbilen,vedaldi}@robots.ox.ac.uk

[2] School of Informatics
University of Edinburgh

## Abstract

There is a growing interest in learning data representations that work well for many different types of problems and data. In this paper, we look in particular at the task of learning a single visual representation that can be successfully utilized in the analysis of very different types of images, from dog breeds to stop signs and digits. Inspired by recent work on learning networks that predict the parameters of another, we develop a tunable deep network architecture that, by means of adapter residual modules, can be steered on the fly to diverse visual domains. Our method achieves a high degree of parameter sharing while maintaining or even improving the accuracy of domain-specific representations. We also introduce the *Visual Decathlon Challenge*, a benchmark that evaluates the ability of representations to capture simultaneously ten very different visual domains and measures their ability to perform well uniformly.

## 1   Introduction

While research in machine learning is often directed at improving the performance of algorithms on specific tasks, there is a growing interest in developing methods that can tackle a large variety of different problems within a single model. In the case of perception, there are two distinct aspects of this challenge. The first one is to extract from a given image diverse information, such as image-level labels, semantic segments, object bounding boxes, object contours, occluding boundaries, vanishing points, etc. The second aspect is to model simultaneously many different visual domains, such as Internet images, characters, glyph, animal breeds, sketches, galaxies, planktons, etc (fig. 1).

In this work we explore the second challenge and look at how deep learning techniques can be used to learn *universal representations* [5], *i.e.* feature extractors that can work well in several different image domains. We refer to this problem as *multiple-domain learning* to distinguish it from the more generic multiple-task learning.

Multiple-domain learning contains in turn two sub-challenges. The first one is to develop algorithms that can *learn well from many domains*. If domains are learned sequentially, but this is not a requirement, this is reminiscent of *domain adaptation*. However, there are two important differences. First, in standard domain adaptation (*e.g.* [9]) the content of the images (e.g. "telephone") remains the same, and it is only the *style* of the images that changes (e.g. real life vs gallery image). Instead in our case a domain shift changes both style and content. Secondly, the difficulty is not just to adapt the model from one domain to another, but to do so while making sure that it still performs well on the original domain, *i.e.* to *learn without forgetting* [21].

The second challenge of multiple-domain learning, and our main concern in this paper, is to construct models that can represent compactly all the domains. Intuitively, even though images in different domains may look quite different (*e.g.* glyph *vs.* cats), low and mid-level visual primitives may still

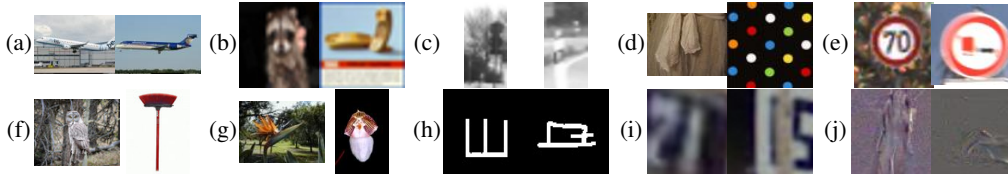

Figure 1: **Visual Decathlon.** We explore deep architectures that can learn simultaneously different tasks from very different visual domains. We experiment with ten representative ones: (a) Aircraft, (b) CIFAR-100, (c) Daimler Pedestrians, (d) Describable Textures, (e) German Traffic Signs, (f) ILSVRC (ImageNet) 2012, (g) VGG-Flowers, (h) OmniGlot, (i) SVHN, (j) UCF101 Dynamic Images.

be largely shareable. Sharing knowledge between domains should allow to learn compact multivalent representations. Provided that sufficient synergies between domains exist, multivalent representations may even work better than models trained individually on each domain (for a given amount of training data).

The primary contribution of this paper (section 3) is to introduce a design for **multivalent neural network architectures for multiple-domain learning** (section 3 fig. 2). The key idea is reconfigure a deep neural network on the fly to work on different domains as needed. Our construction is based on recent learning-to-learn methods that showed how the parameters of a deep network can be predicted from another [2, 16]. We show that these formulations are equivalent to packing the adaptation parameters in convolutional layers added to the network (section 3). The layers in the resulting parametric network are either domain-agnostic, hence shared between domains, or domain-specific, hence parametric. The domain-specific layers are changed based on the ground-truth domain of the input image, or based on an estimate of the latter obtained from an auxiliary network. In the latter configuration, our architecture is analogous to the *learnet* of [2].

Based on such general observations, we introduce in particular a **residual adapter module** and use it to parameterize the standard residual network architecture of [13]. The adapters contain a small fraction of the model parameters (less than 10%) enabling a high-degree of parameter sharing between domains. A similar architecture was concurrently proposed in [31], which also results in the possibility of learning new domains sequentially without forgetting. However, we also show a specific advantage of the residual adapter modules: the ability to modulate adaptation based on the size of the target dataset.

Our proposed architectures are thoroughly evaluated empirically (section 5). To this end, our second contribution is to introduce the **visual decathlon challenge** (fig. 1 and section 4), a new benchmark for multiple-domain learning in image recognition. The challenge consists in performing well simultaneously on ten very different visual classification problems, from ImageNet and SVHN to action classification and describable texture recognition. The evaluation metric, also inspired by the decathlon discipline, rewards models that perform better than strong baselines on all the domains simultaneously. A summary of our finding is contained in section 6.

## 2   Related Work

Our work touches on multi-task learning, learning without forgetting, domain adaptation, and other areas. However, our multiple-domain setup differs in ways that make most of the existing approaches not directly applicable to our problem.

**Multi-task learning (MTL)** looks at developing models that can address different tasks, such as detecting objects and segmenting images, while sharing information and computation among them. Earlier examples of this paradigm have focused on kernel methods [10, 1] and deep neural network (DNN) models [6]. In DNNs, a standard approach [6] is to share earlier layers of the network, training the tasks jointly by means of back-propagation. Caruana [6] shows that sharing network parameters between tasks is beneficial also as a form of regularization, putting additional constraints on the learned representation and thus improving it.

MTL in DNNs has been applied to various problems ranging from natural language processing [8, 22], speech recognition [14] to computer vision [41, 42, 4]. Collobert *et al.* [8] show that semi-supervised learning and multi-task learning can be combined in a DNN model to solve several language processing prediction tasks such as part-of-speech tags, chunks, named entity tags and semantic

roles. Huang *et al.* [14] propose a shared multilingual DNN which shares hidden layers across many languages. Liu *et al.* [22] combine multiple-domain classification and information retrieval for ranking web search with a DNN. Multi-task DNN models are also reported to achieve performance gains in computer vision problems such as object tracking [41], facial-landmark detection [42], object and part detection [4], a collection of low-level and high-level vision tasks [18]. The main focus of these works is learning a diverse set of tasks in the same visual domain. In contrast, our paper focuses on learning a representation from a diverse set of domains.

Our investigation is related to the recent paper of [5], which studied the "size" of the union of different visual domains measured in terms of the capacity of the model required to learn it. The authors propose to absorb different domain in a single neural network by tuning certain parameters in batch and instance normalization layers throughout the architecture; we show that our residual adapter modules, which include the latter as a special case, lead to far superior results.

**Life-long learning.** A particularly important aspect of MTL is the ability of learning multiple tasks sequentially, as in Never Ending Learning [25] and Life-long Learning [38]. Sequential learning typically suffers in fact from forgetting the older tasks, a phenomenon aptly referred to as "catastrophic forgetting" in [11]. Recent work in life-long learning try to address forgetting in two ways. The first one [37, 33] is to freeze the network parameters for the old tasks and learn a new task by adding extra parameters. The second one aims at preserving knowledge of the old tasks by retaining the response of the original network on the new task [21, 30], or by keeping the network parameters of the new task close to the original ones [17]. Our method can be considered as a hybrid of these two approaches, as it can be used to retain the knowledge of previous tasks exactly, while adding a small number of extra parameters for the new tasks.

**Transfer learning.** Sometimes one is interested in maximizing the performance of a model on a target domain. In this case, sequential learning can be used as a form of *initialization*[29]. This is very common in visual recognition, where most DNN are initialize on the ImageNet dataset and then *fine-tuned* on a target domain and task. Note, however, that this typically results in forgetting the original domain, a fact that we confirm in the experiments.

**Domain adaptation.** When domains are learned sequentially, our work can be related to domain adaptation. There is a vast literature in domain adaptation, including recent contributions in deep learning such as [12, 39] based on the idea of minimizing domain discrepancy. Long *et al.* [23] propose a deep network architecture for domain adaptation that can jointly learn adaptive classifiers and transferable features from labeled data in the source domain and unlabeled data in the target domain. There are two important differences with our work: First, in these cases different domains contain the same objects and is only the visual style that changes (*e.g.* webcam *vs.* DSLR), whereas in our case the object themselves change. Secondly, domain adaptation is a form of transfer learning, and, as the latter, is concerned with maximizing the performance on the target domain reagardless of potential forgetting.

## 3 Method

Our primary goal is to develop neural network architectures that can work well in a multiple-domain setting. Modern neural networks such as *residual networks* (ResNet [13]) are known to have very high capacity, and are therefore good candidates to learn from diverse data sources. Furthermore, even when domains look fairly different, they may still share a significant amount of low and mid-level visual patterns. Nevertheless, we show in the experiments (section 5) that learning a ResNet (or a similar model) directly from multiple domains may still not perform well.

In order to address this problem, we consider a *compact parametric family* of neural networks $\phi_\alpha : \mathcal{X} \to V$ indexed by parameters $\alpha$. Concretely, $\mathcal{X} \subset \mathbb{R}^{H \times W \times 3}$ can be a space of RGB images and $V = \mathbb{R}^{H_v \times W_v \times C_v}$ a space of feature tensors. $\phi_\alpha$ can then be obtained by taking all but the last classification layer of a standard ResNet model. The parametric feature extractors $\phi_\alpha$ is then used to construct predictors for each domain $d$ as $\Phi_d = \psi_d \circ \phi_{\alpha_d}$, where $\alpha_d$ are domain-specific parameters and $\psi_d(v) = \text{softmax}(W_d v)$ is a domain-specific linear classifier $V \to \mathcal{Y}_d$ mapping features to image labels.

If $\alpha$ comprises *all* the parameters of the feature extractor $\phi_\alpha$, this approach reduces to learning independent models for each domain. On the contrary, our goal is to maximize parameter sharing, which we do below by introducing certain network parametrizations.

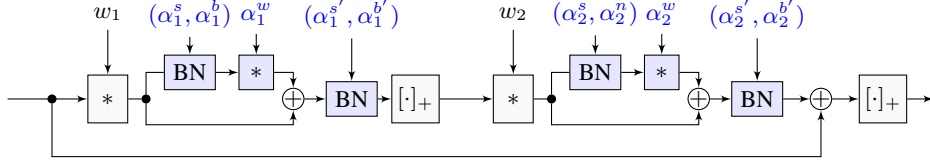

Figure 2: **Residual adapter modules.** The figure shows a standard residual module with the inclusion of adapter modules (in blue). The filter coefficients $(w_1, w_2)$ are domain-agnostic and contains the vast majority of the model parameters; $(\alpha_1, \alpha_2)$ contain instead a small number of domain-specific parameters.

## 3.1 Learning to learn and filter prediction

The problem of adapting a neural network dynamically to variations of the input data is similar to the one found in recent approaches to *learning to learn*. A few authors [34, 16, 2], in particular, have proposed to learn neural networks that predict, in a data-dependent manner, the parameters of another. Formally, we can write $\alpha_d = Ae_{d_x}$ where $e_{d_x}$ is the indicator vector of the domain $d_x$ of image $x$ and $A$ is a matrix whose columns are the parameter vectors $\alpha_d$. As shown later, it is often easy to construct an auxiliary network that can predict $d$ from $x$, so that the parameter $\alpha = \psi(x)$ can also be expressed as the output of a neural network. If $d$ is known, then $\psi(x, d) = \alpha_d$ as before, and if not $\psi$ can be constructed as suggested above or from scratch as done in [2].

The result of this construction is a network $\phi_{\psi(x)}(x)$ whose parameters are predicted by a second network $\psi(x)$. As noted in [2], while this construction is conceptually simple, its implementation is more subtle. Recall that the parameters $w$ of a deep convolutional neural network consist primarily of the coefficients of the linear filters in the convolutional layers. If $w = \alpha$, then $\alpha = \psi(x)$ would need to predict millions of parameters (or to learn independent models when $d$ is observed). The solution of [2] is to use a low-rank decomposition of the filters, where $w = \pi(w_0, \alpha)$ is a function of a filter basis $w_0$ and $\alpha$ is a small set of tunable parameters.

Here we build on the same idea, with some important extensions. First, we note that linearly parametrizing a filter bank is the same as *introducing a new, intermediate convolutional layer* in the network. Specifically, let $F_k \in \mathbb{R}^{H_f \times W_f \times C_f}$ be a basis of $K$ filters of size $H_f \times W_f$ operating on $C_f$ input feature channels. Given parameters $[\alpha_{tk}] \in \mathbb{R}^{T \times K}$, we can express a bank of $T$ filters as linear combinations $G_t = \sum_{k=1}^{K} \alpha_{tk} F_k$. Applying the bank to a tensor $x$ and using associativity and linearity of convolution results in $G * x = \sum_{k=1}^{K} \alpha_{:k}(F_k * x) = \alpha * F * x$ where we interpreted $\alpha$ as a $1 \times 1 \times T \times K$ filter bank. While [2] used a slightly different low-rank filter decomposition, their parametrization can also be seen as introducing additional filtering layers in the network.

An advantage of this parametrization is that it results in a useful decomposition, where part of the convolutional layers contain the domain-agnostic parameters $F$ and the others contain the domain-specific ones $\alpha_d$. As discussed in section 5, this is particularly useful to address the forgetting problem. In the next section we refine these ideas to obtain an effective parametrization of residual networks.

## 3.2 Residual adapter modules

As an example of parametric network, we propose to modify a standard residual network. Recall that a ResNet is a chain $g_m \circ \cdots \circ g_1$ of residual modules $g_t$. In the simplest variant of the model, each residual module $g$ takes as input a tensor $\mathbb{R}^{H \times W \times C}$ and produces as output a tensor of the same size using $g(x; w) = x + ((w_2 * \cdot) \circ [\cdot]_+ \circ (w_1 * \cdot))(x)$. Here $w_1$ and $w_2$ are the coefficients of banks of small linear filters, $[z]_+ = \max\{0, z\}$ is the ReLU operator, $w * z$ is the convolution of $z$ by the filter bank $w$, and $\circ$ denotes function composition. Note that, for the addition to make sense, filters must be configured such that the dimensions of the output of the last bank are the same as $x$.

Our goal is to parametrize the ResNet module. As suggested in the previous section, rather than changing the filter coefficients directly, we introduce additional parametric convolutional layers. In fact, we go one step beyond and make them small residual modules in their own right and call them

*residual adapter modules* (blue blocks in fig. 2). These modules have the form:

$$g(x; \alpha) = x + \alpha * x.$$

In order to limit the number of domain-specific parameters, $\alpha$ is selected to be a bank of $1 \times 1$ filters.

A major advantage of adopting a residual architecture for the adapter modules is that the adapters reduce to the identity function when their coefficients are zero. When learning the adapters on small domains, this provides a simple way of controlling over-fitting, resulting in substantially improved performance in some cases.

**Batch normalization and scaling.** Batch Normalization (BN) [15] is an important part of very deep neural networks. This module is usually inserted after convolutional layers in order to normalize their outputs and facilitate learning (fig. 2). The normalization operation is followed by rescaling and shift operations $s \odot x + b$, where $(s, b)$ are learnable parameters. In our architecture, we incorporate the BN layers into the adapter modules (fig. 2). Furthermore, we add a BN module right before the adapter convolution layer.[1] Note that the BN scale and bias parameters are also dataset-dependent – as noted in the experiments, this alone provides a certain degree of model adaptation.

**Domain-agnostic vs domain-specific parameters.** If the residual module of fig. 2 is configured to process an input tensor with $C$ feature channels, and if the domain-agnostic filters $w_1, w_2$ are of size $h \times h \times C$, then the model has $2(h^2 C^2 + hC)$ domain-agnostic parameters (including biases in the convolutional layers) and $2(C^2 + 5C)$ domain-specific parameters.[2] Hence, there are approximately $h^2$ more domain-agnostic parameters than domain specific ones (usually $h^2 = 9$).

### 3.3 Sequential learning and avoiding forgetting

While in this paper we are not concerned with sequential learning, we have found it to be a good strategy to bootstrap a model when a large number of domains have to be learned. However, the most popular approach to sequential learning, *fine-tuning* (section 2), is often a poor choice for learning shared representations as it tends to quickly forget the original tasks.

The challenge in learning without forgetting is to maintain information about older tasks as new ones are learned (section 2). With respect to forgetting, our adapter modules are similar to the tower model [33] as they preserve the original model exactly: one can pre-train the domain-agnostic parameters $w$ on a large domain such as ImageNet, and then fine-tune only the domain-specific parameters $\alpha_d$ for each new domain. Like the tower method, this preserves the original task exactly, but it is far less expensive as it does not require to introduce new feature channels for each new domain (a quadratic cost). Furthermore, the residual modules naturally reduce to the identity function when sufficient shrinking regularization is applied to the adapter weights $\alpha^w$. This allows the adapter to be tuned depending on the availability of data for a target domain, sometimes significantly reducing overfitting.

## 4   Visual decathlon

In this section we introduce a new benchmark, called *visual decathlon*, to evaluate the performance of algorithms in multiple-domain learning. The goal of the benchmark is to assess whether a method can successfully learn to perform well in several different domains at the same time. We do so by choosing ten representative visual domains, from Internet images to characters, as well as by selecting an evaluation metric that rewards performing well on all tasks.

**Datasets.** The decathlon challenge combines ten well-known datasets from multiple visual domains: **FGVC-Aircraft Benchmark** [24] contains 10,000 images of aircraft, with 100 images for each of 100 different aircraft model variants such as Boeing 737-400, Airbus A310. **CIFAR100** [19] contains 60,000 $32 \times 32$ colour images for 100 object categories. **Daimler Mono Pedestrian Classification Benchmark (DPed)** [26] consists of 50,000 grayscale pedestrian and non-pedestrian images, cropped and resized to $18 \times 36$ pixels. **Describable Texture Dataset (DTD)** [7] is a texture database, consisting of 5640 images, organized according to a list of 47 terms (categories) such as bubbly, cracked,

marbled. **The German Traffic Sign Recognition (GTSR) Benchmark** [36] contains cropped images for 43 common traffic sign categories in different image resolutions. **Flowers102** [28] is a fine-grained classification task which contains 102 flower categories from the UK, each consisting of between 40 and 258 images. **ILSVRC12 (ImNet)** [32] is the largest dataset in our benchmark contains 1000 categories and 1.2 million images. **Omniglot** [20] consists of 1623 different handwritten characters from 50 different alphabets. Although the dataset is designed for one-shot learning, we use the dataset for standard multi-class classification task and include all the character categories in train and test splits. **The Street View House Numbers (SVHN)** [27] is a real-world digit recognition dataset with around 70,000 $32 \times 32$ images. **UCF101** [35] is an action recognition dataset of realistic human action videos, collected from YouTube. It contains 13,320 videos for 101 action categories. In order to make this dataset compatible with our benchmark, we convert the videos into images by using the Dynamic Image encoding of [3] which summarizes each video into an image based on a ranking principle.

**Challenge and evaluation.** Each dataset $\mathcal{D}_d$, $d = 1, \ldots, 10$ is formed of pairs $(x, y) \in \mathcal{D}_d$ where $x$ is an image and $y \in \{1, \ldots, C_d\} = \mathcal{Y}_d$ is a label. For each dataset, we specify a training, validation and test subsets. The goal is to train the best possible model to address all ten classification tasks using only the provided training and validation data (no external data is allowed). A model $\Phi$ is evaluated on the test data, where, given an image $x$ and its ground-truth domain $d_x$ label, it has to predict the corresponding label $y = \Phi(x, d_x) \in \mathcal{Y}_d$.

Performance is measured in terms of a single scalar score $S$ determined as in the *decathlon* discipline. Performing well at this metric requires algorithms to perform well in all tasks, compared to a minimum level of baseline performance for each. In detail, $S$ is computed as follows:

$$S = \sum_{d=1}^{10} \alpha_d \max\{0, E_d^{\max} - E_d\}^{\gamma_d}, \qquad E_d = \frac{1}{|\mathcal{D}_d^{\text{test}}|} \sum_{(x,y) \in \mathcal{D}_d^{\text{test}}} \mathbf{1}_{\{y \neq \Phi(x,d)\}}. \qquad (1)$$

where $E_d$ is the average test error for each domain. $E_d^{\max}$ the baseline error (section 5), above which no points are scored. The exponent $\gamma_d \geq 1$ rewards more reductions of the classification error as this becomes close to zero and is set to $\gamma_d = 2$ for all domains. The coefficient $\alpha_d$ is set to $1,000 \, (E_d^{\max})^{-\gamma_d}$ so that a perfect result receives a score of 1,000 (10,000 in total).

**Data preprocessing.** Different domains contain a different set of image classes as well as a different number of images. In order to reduce the computational burden, all images have been resized isotropically to have a shorter side of 72 pixels. For some datasets such as ImageNet, this is a substantial reduction in resolution which makes training models much faster (but still sufficient to obtain excellent classification results with baseline models). For the datasets for which there exists training, validation, and test subsets, we keep the original splits. For the rest, we use 60%, 20% and 20% of the data for training, validation, and test respectively. For the ILSVRC12, since the test labels are not available, we use the original validation subset as the test subset and randomly sample a new validation set from their training split. We are planning to make the data and an evaluation server public soon.

# 5 Experiments

In this section we evaluate our method quantitatively against several baselines (section 5.1), investigate the ability of the proposed techniques to learn models for ten very diverse visual domains.

**Implementation details.** In all experiments we choose to use the powerful ResNets [13] as base architectures due to their remarkable performance. In particular, as a compromise of accuracy and speed, we chose the ResNet28 model [40] which consists of three blocks of four residual units. Each residual unit contains $3 \times 3$ convolutional, BN and ReLU modules (fig. 2). The network accepts $64 \times 64$ images as input, downscales the spatial dimensions by two at each block and ends with a global average pooling and a classifier layer followed by a softmax. We set the number of filters to $64, 128, 256$ for these blocks respectively. Each network is optimized to minimize its cross-entropy loss with stochastic gradient descent. The network is run for 80 epochs and the initial learning rate of $0.1$ is lowered to $0.01$ and then $0.001$ gradually.

| Model | #par. | ImNet | Airc. | C100 | DPed | DTD | GTSR | Flwr | OGlt | SVHN | UCF | mean | S |
|---|---|---|---|---|---|---|---|---|---|---|---|---|---|
| # images | | 1.3m | 7k | 50k | 30k | 4k | 40k | 2k | 26k | 70k | 9k | | |
| Scratch | 10× | 59.87 | 57.10 | 75.73 | 91.20 | 37.77 | 96.55 | 56.30 | 88.74 | 96.63 | 43.27 | 70.32 | 1625 |
| Scratch+ | 11× | 59.67 | 59.59 | 76.08 | 92.45 | 39.63 | 96.90 | 56.66 | 88.74 | 96.78 | 44.17 | 71.07 | 1826 |
| Feature extractor | 1× | 59.67 | 23.31 | 63.11 | 80.33 | 45.37 | 68.16 | 73.69 | 58.79 | 43.54 | 26.80 | 54.28 | 544 |
| Finetune | 10× | 59.87 | 60.34 | 82.12 | 92.82 | 55.53 | 97.53 | 81.41 | 87.69 | 96.55 | 51.20 | 76.51 | 2500 |
| LwF [21] | 10× | 59.87 | 61.15 | 82.23 | 92.34 | 58.83 | 97.57 | 83.05 | 88.08 | 96.10 | 50.04 | 76.93 | 2515 |
| BN adapt. [5] | ∼ 1× | 59.87 | 43.05 | 78.62 | 92.07 | 51.60 | 95.82 | 74.14 | 84.83 | 94.10 | 43.51 | 71.76 | 1363 |
| Res. adapt. | 2× | 59.67 | 56.68 | 81.20 | 93.88 | 50.85 | 97.05 | 66.24 | 89.62 | 96.13 | 47.45 | 73.88 | 2118 |
| Res. adapt. decay | 2× | 59.67 | 61.87 | 81.20 | 93.88 | 57.13 | 97.57 | 81.67 | 89.62 | 96.13 | 50.12 | 76.89 | 2621 |
| Res. adapt. finetune all | 2× | 59.23 | 63.73 | 81.31 | 93.30 | 57.02 | 97.47 | 83.43 | 89.82 | 96.17 | 50.28 | 77.17 | 2643 |
| Res. adapt. dom-pred | 2.5× | 59.18 | 63.52 | 81.12 | 93.29 | 54.93 | 97.20 | 82.29 | 89.82 | 95.99 | 50.10 | 76.74 | 2503 |
| Res. adapt. (large) | ∼ 12× | 67.00 | 67.69 | 84.69 | 94.28 | 59.41 | 97.43 | 84.86 | 89.92 | 96.59 | 52.39 | 79.43 | 3131 |

Table 1: Multiple-domain networks. The figure reports the (top-1) classification accuracy (%) of different models on the decathlon tasks and final decathlon score ($S$). ImageNet is used to prime the network in every case, except for the networks trained from scratch. The model size is the number of parameters w.r.t. the baseline ResNet. The fully-finetuned model, written blue, is used as a baseline to compute the decathlon score.

| Model | Airc. | | C100 | | DPed | | DTD | | GTSR | | Flwr | | OGlt | | SVHN | | UCF | |
|---|---|---|---|---|---|---|---|---|---|---|---|---|---|---|---|---|---|---|
| Finetune | 1.1 | 60.3 | 3.6 | 63.1 | 0.6 | 80.3 | 0.7 | 45.3 | 1.4 | 68.1 | 27.2 | 73.6 | 13.4 | 87.7 | 0.2 | 96.6 | 5.4 | 51.2 |
| LwF [21] high lr | 4.1 | 61.1 | 21.0 | 82.2 | 23.8 | 92.3 | 36.7 | 58.8 | 11.5 | 97.6 | 34.2 | 83.1 | 3.0 | 88.1 | 0.2 | 96.1 | 18.6 | 50.0 |
| LwF [21] low lr | 38.0 | 50.6 | 33.0 | 80.7 | 53.3 | 92.2 | 47.0 | 57.2 | 23.7 | 96.6 | 45.7 | 75.7 | 21.0 | 86.0 | 13.3 | 94.8 | 29.0 | 44.6 |
| Res. adapt. finetune all | 59.2 | 63.7 | 59.2 | 81.3 | 59.2 | 93.3 | 59.2 | 57.0 | 59.2 | 97.5 | 59.2 | 83.4 | 59.2 | 89.8 | 59.2 | 96.1 | 59.2 | 50.3 |

Table 2: Pairwise forgetting. Each pair of numbers report the top-1 accuracy (%) on the old task (ImageNet) and a new target task after the network is fully finetuned on the latter. We also show the performance of LwF when it is finetuned on the new task with a high and low learning rate, trading-off forgetting ImageNet and improving the results on the target domain. By comparison, we show the performance of tuning only the residual adapters, which by construction does not result in any performance loss in ImageNet while still achieving very good performance on each target task.

## 5.1 Results

There are two possible extremes. The first one is to learn ten independent models, one for each dataset, and the second one is to learn a single model where all feature extractor parameters are shared between the ten domains. We evaluate next different approaches to learn such models.

**Pairwise learning.** In the first experiment (table 1), we start by learning a ResNet model on ImageNet, and then use different techniques to extend it to the remaining nine tasks, one at a time. Depending on the method, this may produce an overall model comprising ten ResNet architectures, or just one ResNet with a few domain-specific parameters; thus we also report the total number of parameters used, where 1× is the size of a single ResNet (excluding the last classification layer, which can never be shared).

As baselines, we evaluate four cases: i) learning an individual ResNet model from scratch for each task, ii) freezing all the parameters of the pre-trained network, using the network as feature extractor and only learn a linear classifier, iii) standard finetuning and iv) applying a reimplementation of the LwF technique of [21] that encourages the fine-tuned network to retain the responses of the original ImageNet model while learning the new task.

In terms of accuracy, learning from scratch performs poorly on small target datasets and, by learning 10 independent models, requires 10× parameters in total. Freezing the ImageNet feature extraction is very efficient in terms of parameter sharing (1× parameters in total), preserves the original domain exactly, but generally performs very poorly on the target domain. Full fine-tuning leads to accurate results both for large and small datasets; however, it also forgets the ImageNet domain substantially (table 2), so it still requires learning 10 complete ResNet models for good overall performance.

When LwF is run as intended by the original authors [21], is still leads to a noticeable performance drop on the original task, even when learning just two domains (table 2), particularly if the target domain is very different from ImageNet (*e.g*. Omniglot and SVHN). Still, if one chooses a different trade-off point and allows the method to forget ImageNet more, it can function as a good regularizer that slightly outperforms vanilla fine-tuning overall (but still resulting in a 10× model).

Next, we evaluate the effect of sharing the majority of parameters between tasks, whereas still allowing a small number of domain-specific parameters to change. First, we consider specializing only the BN layer scaling and bias parameters, which is equivalent to the approach of [5]. In this case, less than the $0.1\%$ of the model parameters are domain-specific (for the ten domains, this results in a model with $1.01\times$ parameters overall). Hence the model is very similar to the one with the frozen feature extractor; nevertheless, the performances increase very substantially in most cases (e.g. $23.31\% \rightarrow 43.05\%$ accuracy on Aircraft).

As the next step, we introduce the **residual adapter modules**, which increase by $11\%$ the number of parameters per domain, resulting in a $2\times$ model. In the pre-training phase, we first pretrain on ImageNet the network with the added modules. Then, we freeze the task agnostic parameters and train the task specific parameters on the different datasets. Differently from vanilla fine-tuning, there is no forgetting in this setting. While most of the parameters are shared, our method is either close or better than full fine-tuning. As a further control, we also train 10 models from scratch with the added parameters (denoted as Scratch+), but do not observe any noticeable performance gain in average, demonstrating that parameters sharing is highly beneficial. We also contrast learning the adapter modules with two values of weight decay (0.002 and 0.005) higher than the default 0.0005. These parameters are obtained after a coarse grid search using cross-validation for each dataset. Using higher decay significantly improves the performance on smaller datasets such as Flowers, whereas the smaller decay is best for larger datasets. This shows both the importance and utility of controlling overfitting in the adaptation process. In practice, there is an almost direct correspondence between the size of the data and which one of these values to use. The optimal decay can be selected via validation, but a rough choice can be performed by simply looking at the dataset size.

We also compare to another baseline where we only finetune the last two convolutional layers and freeze the others, which may be thought to be generic. This amounts to having a network with twice the number of total parameters in a vanilla ResNet which is equal to our proposed architecture. This model obtains $64.7\%$ mean accuracy over ten datasets, which is significantly lower than our $73.9\%$, likely due to overfitting (controlling overfitting is one of the advantages of our technique).

Furthermore, we also assess the quality of our adapter without residual connections, which corresponds to the low rank filter parametrization of section 3.1; this approach achieves an accuracy of $70.3\%$, which is worse than our $73.9\%$. We also observe that this configuration requires notably more iterations to converge. Hence, the residual architecture for the adapters results in better performances, better control of overfitting, and a faster convergence.

**End-to-end learning.** So far, we have shown that our method, by learning only the adapter modules for each new domain, does not suffer from forgetting. However, for us sequential learning is just a scalable learning strategy. Here, we also show (table 1) that we can further improve the results by fine-tuning all the parameters of the network end-to-end on the ten tasks. We do so by sampling a batch from each dataset in a round robin fashion, allowing each domain to contribute to the shared parameters. A final pass is done on the adapter modules to take into account the change in the shared parameters.

**Domain prediction.** Up to now we assume that the domain of each image is given during test time for all the methods. If this is unavailable, it can be predicted on the fly by means of a small neural-network predictor. We train a light ResNet, which is composed three stacks of two residual networks, half deep as the original net, obtaining $99.8\%$ accuracy in domain prediction, resulting in a barely noticeable drop in the overall multiple-domain challenge (see Res. adapt dom-pred in table 1). Note that similar performance drop would be observed for the other baselines.

**Decathlon evaluation: overall performance.** While so far we have looked at results on individual domain, the Decathlon score eq. (1) can be used to compare performance overall. As baseline error rates in eq. (1), we double the error rates of the fully finetuned networks on each task. In this manner, this $10\times$ model achieves a score of 2,500 points (over 10,000 possible ones, see eq. (1)). The last column of table 1 reports the scores achieved by the other architectures. As intended, the decathlon score favors the methods that perform well overall, emphasizes their *consistency* rather than just their average accuracy. For instance, although the Res. adapt. model (trained with single decay coefficient for all domains) performs well in terms of average accuracy (73.88%), its decathlon score (2118) is relatively low because the model performs poorly in DTD and Flowers. This also shows that, once the weight decays are configured properly, our model achieves superior performance (2643 points) to all the baselines using only $2\times$ the capacity of a single ResNet.

Finally we show that using a higher capacity ResNet28 (12×, ResNet adapt. (large) in table 1), which is comparable to 10 independent networks, significantly improves our results and outperforms the finetuning baseline by 600 point in decathlon score. As a matter of fact, this model outperforms the state-of-the-art [40] (81.2%) by 3.5 points in CIFAR100. In other cases, our performances are in general in line to current state-of-the-art methods. When this is not the case, this is due to reduced image resolution (ImageNet, Flower) or due to the choice of a specific video representation in UCF (dynamic image).

## 6  Conclusions

As machine learning applications become more advanced and pervasive, building data representations that work well for multiple problems will become increasingly important. In this paper, we have introduced a simple architectural element, the residual adapter module, that allows compressing many visual domains in relatively small residual networks, with substantial parameter sharing between them. We have also shown that they allow addressing the forgetting problem, as well as adapting to target domain for which different amounts of training data are available. Finally, we have introduced a new multi-domain learning challenge, the Visual Decathlon, to allow a systematic comparison of algorithms for multiple-domain learning.

**Acknowledgments:** This work acknowledges the support of Mathworks/DTA DFR02620 and ERC 677195-IDIU.

## Footnotes

[1]While the bias and scale parameters of the latter can be incorporated in the following filter bank, we found it easier to leave them separated from the latter

[2]Including all bias and scaling vectors; $2(C^2 + 3C)$ if these are absorbed in the filter banks when possible.

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
