[Supplementary Material]

# A Decathlon scores

| Model | #par. | ImNet | Airc. | C100 | DPed | DTD | GTSR | Flwr | OGlt | SVHN | UCF | De-cathlon score |
|---|---|---|---|---|---|---|---|---|---|---|---|---|
| Scratch | 10× | 250 | 211 | 103 | 150 | 90 | 91 | 0 | 294 | 261 | 175 | 1625 |
| Scratch+ | 11× | 247 | 241 | 110 | 226 | 103 | 138 | 0 | 294 | 284 | 183 | 1826 |
| Feature extractor | 1× | 247 | 1 | 0 | 0 | 149 | 0 | 85 | 0 | 0 | 62 | 544 |
| Finetune | 10× | 250 | 250 | 250 | 250 | 250 | 250 | 250 | 250 | 250 | 250 | 2500 |
| LwF [1] | 10× | 250 | 260 | 253 | 218 | 288 | 258 | 296 | 266 | 188 | 238 | 2515 |
| BN adapt. | ∼ 1× | 250 | 80 | 162 | 201 | 208 | 24 | 93 | 147 | 21 | 177 | 1363 |
| Res. adapt. | 2× | 247 | 206 | 225 | 329 | 200 | 163 | 8 | 335 | 192 | 213 | 2118 |
| Res. adapt. decay | 2× | 247 | 270 | 225 | 330 | 268 | 258 | 257 | 335 | 192 | 239 | 2621 |
| Res. adapt. finetune all | 2× | 242 | 295 | 228 | 285 | 267 | 237 | 307 | 344 | 197 | 241 | 2643 |
| Res. adapt. dom-pred | 2.5× | 241 | 292 | 223 | 284 | 243 | 188 | 274 | 344 | 175 | 239 | 2503 |
| Res. adapt. (large) | ∼ 12× | 347 | 351 | 327 | 362 | 296 | 231 | 351 | 349 | 255 | 262 | 3131 |

Table 1: Multiple-domain networks. The figure reports the decathlon score of different models on the multiple tasks. ImageNet is used to prime the network in every case, except for the networks trained from scratch. The model size is the number of parameters w.r.t. the baseline ResNet. The fully-finetuned model, written blue, is used as a baseline to compute the decathlon score.

# References

[1] Z. Li and D. Hoiem. Learning without forgetting. In *Proc. ECCV*, pages 614–629, 2016.