[Reviews · NeurIPS 2017]

Reviewer 1



Paper address the problem of learning a joint recognition model across variety of domains and tasks. It develops a tunable deep network architecture that, by means of adapter residual modules, can be steered for a variety of visual domains. Methods archives good performance across variety of domains (10 in total) while having high degree of parameter sharing. Paper also introduces a corresponding challenge benchmark - Decathlon Challenge. The benchmark evaluates the ability of a representation to be applied across 10 very different visual domains.
 Generally the paper is very well written and presents an interesting approach and insights. I particularly applaud drawing references to related works and putting the approach in true context. Additional comments are below. > References Generally the references are good and adequate. The following references are thematically related and should probably be added: Learning Deep Feature Representations with Domain Guided Dropout for Person Re-identification, Tong Xiao, Hongsheng Li, Wanli, Ouyang, Xiaogang Wang, CVPR, 2016. Expanding Object Detector's Horizon: Incremental Learning Framework for Object Detection in Videos, Alina Kuznetsova, Sung Ju Hwang, Bodo Rosenhahn, and Leonid Sigal, CVPR, 2015. > Clarifications Lines 201-202 talk about shrinking regularization. I think this is very interesting and could be a major benefit of the model. However, it is unclear if this is actually tested in practice in experiments. Can you please clarify. Also, authors claim that the proposed model ensures that performance does not degrade in the original domain. This does not actually appears to be the case (or supported experimentally). For example, performance on ImageNet in Table 1 goes from 59.9 to 59.2. This should be discussed and explained. > Experiments In light of the comments above, Table 2 should include the proposed model (or models), e.g., Res. adapt. decay and Res. adapt. finetune all. Finally, explanation of Scratch+ is unclear and should be clarified in Lines 303-305. I read the sentence multiple times and still unsure what was actually done.

Reviewer 2



The paper presents a CNN architecture able to learn over different domains with a limited amount of parameters and without the need to forget previous knowledge. An original ResNet is modified by adding multiple modules composed by residual connections and batch normalization layers: the paper shows that by only learning the parameters of these modules the network is able to produce good performance on different domains. + the paper is easy to read, the objectives as well defined and the method is clearly explained + besides the method, the author introduce a visual decathlon challenge that may be used as reference for other domain generic methods in the future few points that need some discussion: - on page 4, lines 148-155 the paper explains that the idea proposed here is inspired by a previous work where a low rank filter decomposition was used instead of the introduction of separated modules. Despite this practical difference it sounds like it would be possible to test this method on the decathlon challenge, but it is not used here as reference in the expers. - the method has been designed to cope with domains where the samples do not only differ in terms of style but also in their class set. Still nothing avoids to test the method in the easier domain adaptation setting with shared classes across domains. What would happen there? Would it produce better or worse results than sota deep DA methods? - as far as I'm concerned the FT procedure can be done in different ways depending on the number of layers that are kept fixed (frozen) and the number of layers for which the parameters can be updated. Thus I think FT is not necessarily a 10x process in terms of number of parameters with respect to the original ResNet. What would happen by updating less parameters? - In line 337 of page 8 the sota result on CIFAR is mentioned. Why no of the other databases sota results are provided as reference? It could be useful to understand how the method compares with them.

Reviewer 3



This paper explores how to learn feature extractors that support learning in several different image domains. The general idea is to learn a residual adapter module and use it to parameterize the standard residual network architecture. there is a high-degree of parameter sharing between domains. One particular advantage is to enable the learning of new domains sequentially without forgetting the previous learned results. I find the 'avoid forgetting' part of the residual network model to be interesting, but this part is too short and the discussions at a high level, that I cannot figure out how much the 'forgetting' problem is addressed by this model. For example, can there be some theoretical guarantee on how much forgetting is prevented? Can there be some comparison on how much more is gained using the residual network model? The experiments in Section 5 are detailed and well done, but it is a pity that the authors did not give a sequential learning setting similar to lifelong learning, where the learning curve is shown as a function of domains. Some permutation that can result in better gain in learning performance can be discussed as well.